# Cancer Cell Metabolism Reprogramming and Its Potential Implications on Therapy in Squamous Cell Carcinoma of the Head and Neck: A Review

**DOI:** 10.3390/cancers14153560

**Published:** 2022-07-22

**Authors:** Francesco Perri, Giuseppina Della Vittoria Scarpati, Monica Pontone, Maria Luisa Marciano, Alessandro Ottaiano, Marco Cascella, Francesco Sabbatino, Agostino Guida, Mariachiara Santorsola, Piera Maiolino, Ernesta Cavalcanti, Giulia Togo, Franco Ionna, Francesco Caponigro

**Affiliations:** 1Medical and Experimental Head and Neck Oncology Unit, INT IRCSS Foundation G. Pascale, 80131 Naples, Italy; m.pontone@istitutotumori.na.it (M.P.); ml.marciano@istitutotumori.na.it (M.L.M.); f.caponigro@istitutotumori.na.it (F.C.); 2Medical Oncology Unit, Hospital Sir Apicella, Pollena Trocchia, 80040 Naples, Italy; giuseppina.dellavittoria@gmail.com; 3SSD Innovative Therapies for Abdominal metastases, Abdominal Oncology, INT IRCCS Foundation G. Pascale, 80131 Naples, Italy; a.ottaiano@istitutotumori.na.it (A.O.); mariachiara.santorsola@istitutotumori.na.it (M.S.); 4Unit of Anestesiology and Pain Therapy, INT IRCCS Foundation G. Pascale, 80131 Naples, Italy; m.cascella@istitutotumori.na.it; 5Oncology Unit, Department of Medicine, Surgery and Dentistry, University of Salerno, Baronissi, 84081 Salerno, Italy; fsabbatino@unisa.it; 6U.O.C. Odontostomatologia, AORN A. Cardarelli Hospital, 80131 Naples, Italy; agostino.guida@aocardarelli.it; 7Pharmacy Unit, INT IRCCS Foundation G. Pascale, 80131 Naples, Italy; p.maiolino@istitutotumori.na.it; 8Laboratory Medicine, INT IRCCS Foundation G. Pascale, 80131 Naples, Italy; e.cavalcanti@istitutotumori.na.it; 9Maxillofacial Surgery Unit, Department of Neuroscience, Reproductive and Odontostomatological Sciences, University of Naples Federico II, 80131 Naples, Italy; giulia.togo@gmail.com; 10Otolaryngology Unit, INT IRCCS Foundation G. Pascale, 80131 Naples, Italy; f.ionna@istitutotumori.na.it

**Keywords:** head and neck squamous cell carcinoma, metabolic reprogramming, immunosurveillance, Akt, TP53, Warburg effect

## Abstract

**Simple Summary:**

Metabolic reprogramming is a common phenomenon occurring in several solid tumors, among which are head and neck cancers, which allows them to better survive and progress. The main scope of the present work is to integrate the biological mechanisms underlying metabolic reprogramming, which often occurs in cancer cells, with the current therapeutic possibilities in the field of squamous neoplasms of the head and neck. The final goal is to provide a possible therapeutic approach that employs different combinations of chemotherapy, targeted therapy, and immunotherapy on the basis of the biology of the tumor. Counteracting metabolic reprogramming could be an interesting strategy in the near future.

**Abstract:**

Carcinogenesis is a multistep process that consists of the transformation of healthy cells into cancer cells. Such an alteration goes through various stages and is closely linked to random mutations of genes that have a key role in the neoplastic phenotype. During carcinogenesis, cancer cells acquire and exhibit several characteristics including sustaining proliferative signaling, evading growth suppressors, resisting cell death, enabling replicative immortality, inducing angiogenesis, activating invasion and metastasis, and expressing an immune phenotype, which allow them to evade recognition and destruction through cognate immune cells. In addition, cancer cells may acquire the ability to reprogram their metabolism in order to further promote growth, survival, and energy production. This phenomenon, termed metabolic reprogramming, is typical of all solid tumors, including squamous carcinomas of the head and neck (SCCHN). In this review, we analyze the genetic and biological mechanisms underlying metabolic reprogramming of SCCHN, focusing on potential therapeutic strategies that are able to counteract it.

## 1. Introduction

The process of neoplastic transformation normally goes through three distinct phases: initiation, promotion, and progression. In the first phase, a gene mutation predisposes cancer development. During the second phase, accumulation of other genetic mutations allows cells to acquire a cancerous phenotype. Lastly, in the progression phase, the further accumulation of mutations causes genetic heterogeneity, as well as the acquisition of a full malignant phenotype. During this phase, the tumor increases its aggressiveness, circumventing the “barriers” created by the immune system [1,2].

During cancerogenesis, cells acquire distinctive characteristics, which are defined as hallmarks of cancer. They include a limitless replicative potential, sustained angiogenesis, avoidance of apoptosis, self-sufficiency in growth signals, insensitivity to anti-growth signals, and, finally, tissue invasion and metastatic potentiality. Such hallmarks are interrelated and have been extensively studied. More recently, two emerging characteristics have also been added: immune system evasion and energy metabolism reprogramming [3]

The host’s immune system is able to eliminate cancer cells that are constantly emerging throughout life. This “self-defense” mechanism is known as “immunosurveillance” [3,4]. However, during neoplastic progression, through mutations, cancer cells acquire the ability to evade the immune response through various mechanisms such as (i) down-regulation of HLA antigens, (ii) decrease or loss of expression of tumor associated antigens (TAA), and (iii) production of immunosuppressive cytokines (i.e., TGF-beta and IL-10) [4].

In addition, tumor cells might become capable of reprogramming their metabolism, acquiring the ability to survive in difficult conditions. Metabolic reprogramming grants the increased energy request due to continuous growth, rapid proliferation, and other characteristics typical of neoplastic cells. This is a crucial evolutionary advantage of neoplastic cells.

The most significant metabolic pathways involved in this adaptation include anaerobic glycolysis, glutaminolysis, and mitochondrial biogenesis and activities [5].

In this review, we aim to describe the role of the pathways involved metabolic reprogramming in squamous cell carcinoma of the head and neck (SCCHN), and to illustrate the genetic/epigenetic changes underlying metabolic reprogramming pathway activation in order to hypothesize novel potential therapeutic strategies. The main metabolic changes occurring during the neoplastic progression of SCCHN will also be analyzed and described, discussing their impact on cancer aggressiveness. 

## 2. Metabolic Shift in Cancer Cells

One of the best-known metabolic changes in solid tumors is the so-called “Warburg effect”, first described by Otto H. Warburg in 1924. He postulated that the driver of tumorigenesis is an insufficient cellular respiration caused insult to the mitochondria, which in turn causes an enhanced accelerated anaerobic conversion of glucose to lactate by non-oxidative breakdown (a process called anaerobic glycolysis), even in the presence of abundant oxygen. Otto H. Warburg hypothesized that this effect in cancer cells was associated with damage in the cytochromes. Cytochromes are involved in oxidative phosphorylation activities [3,6]. Later, Chance and collaborators showed that cytochromes in tumors cells, even in the presence of enhanced anaerobic glycolysis, were intact and functional [6], thus arguing the hypothesis of mitochondrial dysfunction. In addition, several lines of evidence indicate that glycolysis in tumors can be upregulated without mitochondrial dysfunction or oxidative phosphorylation (OxPhos) alterations. As a result, the Warburg effect cannot be considered as a normal adaptation of tumor cells to hypoxia, a characteristic that is present in almost all solid tumors. This effect is a crucial phenomenon for neoplastic development and progression, appearing early during carcinogenesis [6].

In addition, this effect is not a prerogative of cancer cells, as all rapidly proliferating cells, such as pluripotent stem cells, immune cells, and endothelial cells might also show a switch to Warburg-type glucose metabolism [7,8]. The prevalence of anaerobic glycolysis instead of OxPhos may confer proliferative advantages to cells. Anaerobic glycolysis is a faster process than OxPhos, and despite having a lower energy yield (two molecules of ATP for each molecule of glucose used), it favors the immediate availability of energy (in the form of ATP). The shift towards glycolysis allows for a further shift towards other metabolic pathways that, in some way, privilege proliferation to quiescent cells.

Anaerobic glycolysis includes 10 biochemical steps, all of which are upregulated in both cancer and proliferating cells. Key transcriptional activators of anaerobic glycolysis include the “high-affinity” glucose transporter GLUT1, the lactate exit transporter monocarboxylate transporter 4 (MCT4), glycolytic enzymes (hexokinase 2, phosphofructokinase 1, and enolase 1), and the low-activity pyruvate kinase M2 (PKM2). During anaerobic glycolysis, the product of glucose metabolism, namely pyruvate, is not transferred into the mitochondria (for subsequent Krebs cycle), but is converted into lactate. Lactate is exported into the extracellular space. In addition, during anaerobic glycolysis, some intermediate metabolites accumulate and, in turn, are used to accelerate and upregulate other metabolic pathways such as those involved in the biosynthesis of fatty acids and phospholipids, as well as of proteins, or in the pentose phosphate (PPP) pathway, leading to an increase in the production of intracellular NADPH [9,10,11]. As a result, the immediate availability of energy (in the form of ATP) in proliferating cells, including cancer cells, allows them to regenerate the plasma membranes and to increase the synthesis of proteins and nucleotides, which are essential components needed to complete DNA duplication and mitosis.

In order to maintain energy homeostasis, in addition to glucose, cancer cells can also be utilized as a substrate—another key metabolite, namely glutamine. In cancer cells, glutamine metabolism surpasses any other non-essential amino acid metabolism. It plays an important role in anaplerosis (the replenishment of Krebs cycle metabolites) and NADPH regeneration through lactate production and through restoration of the reduced glutathione pool. Glutamine can also act as a nitrogen donor for the synthesis of purine and pyrimidine nucleotides, amino acids, and other metabolites. Finally, glutamine catabolism results in the production of lactate, alanine, and ammonia, thus maintaining non-essential amino acid pools [12,13,14,15,16]. An increased production of NADPH is essential for the biosynthesis of fatty acids and serves as the ultimate donor of reductive power for detoxifying the cell from the enormous amount of reactive oxygen species (ROS) that accumulate as a result of the high proliferation rate. On the other hand, lactate, the final product of glycolysis, is secreted by cancer cells in the tumor microenvironment, which has an immunosuppressive effect by stimulating M2 macrophage activation. M2 macrophages reduce adaptive Th1 immunity by inhibiting Th1 cytokine production, such as the production of IL-12, IL-2, and IFN gamma. Moreover, M2 macrophages promote angiogenesis and tissue remodelling by releasing TGF-beta in the tumor microenvironment. As a result, the increased production of lactate suppresses the host anti-cancer immune response, leading to tumor progression in solid tumors, including SCCHN [17].

In conclusion, cancer cells often, but not always, show a metabolic shift that favors some reactions rather than others in order to support their growth, proliferation, and immunoescape. The differential metabolic features of normal and cancer cells are shown in Figure 1.

## 3. Intracellular Pathways Involved in Metabolic Reprogramming

Cancer cell metabolic reprogramming is associated with both genetic and epigenetic alterations, which induce the dysregulation of many crucial intracellular/extracellular pathways. Well defined mechanisms, acting independently or in cooperation by sustaining anaerobic glycolysis, the synthesis of fatty acids, nucleotides or proteins, and PPP pathway activation include (i) activation of HIF-1 (hypoxia inducible factor 1), (ii) activation of oncogenes such as cMyc, mammalian target of rapamycin complex 1 (mTORC1), Akt, and K-Ras; (iii) inactivation of tumor suppressor genes such as TP53 and PTEN; and (iv) upregulation of the receptor tyrosine kinase PI3K–Akt–mTORC1 signaling pathway [18,19,20,21,22].

### 3.1. PI3K-Akt-mTOR Pathway

The PI3K−Akt−mTOR pathway is one of the most altered pathways in cancers, including SCCHN. Importantly, PI3KCA is one of the most frequently mutated genes in HPV (human papilloma virus)-related SCCHN.

Akt, also known as Protein Kinase B, is a serine threonine kinase acting upon several intracellular downstream effectors. It is regulated by “growth signals” and is involved in various diseases, including cancer. Akt is activated in response to tyrosine kinase receptor (TRKs) activation. Once the “growth ligand” links the TRK, it induces a conformational change in the receptor’s structure, which in turn recruits a protein called PI3K (Phosphatidil Inositole 3 Kinase) [23]. PI3K converts phosphatidylinositol (4,5)-bisphosphate (PIP2) in phosphatidylinositol (3,4,5)-trisphosphate (PIP3). PIP3 in turn activates PDK1 (phosphoinositide dependent kinase 1). PDK1 phosphorylates and activates Akt directly or through the activation of mTORCH2 [24] (mammalian target of rapamycin complex 2); once activated, Akt interacts with its various downstream substrates, including AS160 (Akt substrate of 160 kDa). AS160 negatively regulates the membrane translocation of the most active and important glucose receptor, GLUT4. Akt activation inhibits AS160 function, thus facilitating GLUT4 activation. As a result, GLUT4 increases the uptake of glucose by cancer cells and its consequent availability for the glycolysis [25]. In addition, Akt activation inhibits TSC1 (hamartin–tuberin complex) function. The TSC1–TSC2 (tuberous sclerosis complex 1−2) complex, through its GAP (GTPase-activating protein) activity mediated by the small G-protein Rheb (Ras homologue enriched in brain), down-regulates mTORC1 activity^26^. mTORC1 activates P60S6K (Ribosomal protein S6 kinase beta 1), an important enzyme involved in protein synthesis. As a result, inhibition of TSC1−TSC2 by Akt stimulates proteins synthesis. Lastly, Akt phosphorylates and up-regulates the enzyme ATP cytrate lyase, which is crucial for fatty acid synthesis [26]. A scheme depicting the above-mentioned changes is shown in Figure 2.

In conclusion, Akt up-regulation strongly stimulates the uptake and utilization of glucose through glycolisis, increases fatty acid synthesis, and upregulates protein synthesis, as well as stimulating cell proliferation and acting on different tumor suppressor genes such as FOXO (Forkhead box O3) [26,27].

### 3.2. TP53

P53 is an enzyme that acts as a transcription factor encoded by the TP53 gene. It needs to tetramerize in order to perform its function and becomes active following damage to the DNA. P53 binds some promoter sequences on the DNA and upregulates the genes involved in cell cycle arrest (through the P21 protein), DNA damage repair (homologous recombination), and in case of too extensive DNA damage in apoptosis induction (action on PUMA (p53 upregulated modulator of apoptosis) genes) [28].

Besides these well-defined functions of p53, more recently, its impact on cell metabolism has been also described.

P53 has been shown to regulate glycolysis, PPP, mitochondrial oxidative phosphorylation, lipid and nucleotide metabolism, and cell response to oxidative stress. Under normal conditions, wild type p53 upregulates OxoPhos and down-regulates both glycolysis and glucose transport within the cell (via the GLUT4 receptor). Furthermore, wild type p53 down-regulates the enzyme glucose-6-phosphate dehydrogenase (G6PD) by inhibiting PPP. Finally, wild type p53 stimulates the beta-oxidation of fatty acids by inhibiting their biosynthesis [28,29].

P53 is the most mutated gene in cancers, including SCCHN. Mutations of P53 are described in about 50% of SCCHN cases. However, only 15% of mutations are “disruptive”, namely they are accompanied by the transduction of a non-functioning protein. On the other hand, the majority of TP53 mutations are “missense”, causing the transduction of a p53 mutated that performs the opposite functions compared with the wild type protein. Specifically, missense p53 mutated down regulates OxoPhos and the beta oxidation of fatty acids and strongly stimulates anaerobic glycolysis, PPP, and fatty acid biosynthesis. As a result, TP53 mutations can induce metabolic reprograming in cancers, including SCCHN [30,31].

### 3.3. HIF-1 Alpha

Tumor hypoxia appears to be closely associated with tumor dissemination, malignant progression, and resistance to radiotherapy. HIF-1 alpha is a transcription factor that is hydroxylated under normal oxygenation conditions, which then becomes susceptible to ubiquitination and then to degradation. In contrast, in hypoxic conditions, HIF-1 alpha is stabilized, upregulated, and stimulates the transcription of the genes involved in angiogenesis and anaerobic glycolysis by binding to the promoter sequences on the DNA. HIF-1alpha is shown to be overexpressed in SCCHN. In addition, its level of expression has both predictive and prognostic significance for SCCHN patients undergoing radiotherapy and chemotherapy. Upregulation of HIF-1 alpha is rarely reported to be caused by gene mutations with “gain of function”, while more frequently, the “disruption” of the pathway is caused by HIF-1alpha polymorphisms. The result in both cases is an up regulation of glycolysis and therefore of the Warburg effect [32,33,34].

### 3.4. PTEN

PTEN is a tumor suppressor gene, which acts as a negative regulator of the PI3K/Akt/mTOR pathway. Specifically, PTEN inhibits PI3K activation. As a result, mutations causing loss-of-function induce Akt up-regulation as well as up-regulation of all of the down-stream pathway components. Loss of PTEN expression is described in about 30% of oral cavity cancers, and loss of heterozygosity (LOH) is identified in 40% of SCCHNs. As a result, AKT activation results in an up-regulation of glycolysis, PPP, biosynthesis of fatty acids, nucleotides, and proteins favoring metabolic reprogramming [35,36].

### 3.5. c-Myc

MYC oncogene contributes to the genesis of many human cancers. In SCCHN, a small percentage of c-MYC alterations have been described [37,38]. It encodes a transcription factor c-Myc, which links altered cellular metabolism to tumorigenesis. c-MYC regulates the genes involved in the regulation of glucose and glutamine metabolism. Studies have demonstrated that MYC can directly bind to the promoters of thousands of genes, including HIF-1 alpha. c-MYC can cooperate with HIF-1 by inducing a transcriptional program for hypoxic adaptation. Specifically, c-MYC can either directly regulate glycolytic genes including lactate dehydrogenase A (LDHA), or indirectly modulate glutamine metabolism by activating the repression of microRNAs miR-23a/b to increase the glutaminase (GLS) protein expression. c-MYC can also stimulate glutamine metabolism through regulation of the transporters (SLC1A5) and glutaminase (GLS). Glutamine is shown to be converted to α-ketoglutarate (α-KG) for catabolism to malate through the tricarboxylic acid (TCA) cycle, which is transported into the cytoplasm and is converted to pyruvate and then to lactate (glutaminolysis). In conclusion, c-MYC stimulates both glycolysis and glutamine catabolism with the aim to produce energy.

## 4. Interplay between Metabolic Shift and Immune Response

The interaction between tumor cell metabolic reprogramming and immunosurveillance can be defined as “immune metabolism”. As we previously discussed, metabolic reprogramming negatively influences the immune response to cancer. 

### 4.1. Immune Surveillance

The phenomenon of immune surveillance represents a powerful weapon against the outbreak of tumor growth, allowing the immune system to eliminate new cancer cells. Immune surveillance is mainly granted by two types of immune cells, namely cytotoxic T lymphocytes (CTLs) and Natural Killer (NK) cells [39]. Both continuously circulate in the peripheral blood and lymphoid and non-lymphoid tissues, including cancer tissues. Altered cancer cells are firstly identified and ultimately destroyed by several mechanisms, including the release of lytic enzymes such as perforins and granzymes, or through the interaction between FAS ligand and FAS receptors expressed in immune and cancer cells, respectively. NK cells use various mechanisms for identifying tumor cells. One of the major mechanisms includes down-regulation of HLA class I antigens, a frequent event in cancer cells. On the other hand, CTLs recognize TAAs expressed in the cell membrane of tumor cells in the context of HLA class I antigen expression. As a result, both NK and CTLs are involved in the elimination of neoplastic clones [40,41,42,43]. Nevertheless, during cancer progression, cancer cells develop mechanisms able to circumvent immune surveillance and become resistant to immune response, a phenomenon defined as immune escape. The acquisition of this phenotype by cancer cells is caused by the genetic instability mediated by the accumulation of random mutations. As a result, cancer cells become “invisible” to the cells of the immune system through down-regulation of HLA class I antigens or TAAs, and through the production of immune suppressive cytokines such as TGF-beta, IL-4, and IL-10.

### 4.2. Metabolic Reprogramming and Immune Suppression

Several cells of the immune system, likely cancer cells, might implement a metabolic reprograming. Both anaerobic glycolysis and glutamine metabolizing are often used instead of OxoPhos in various immune cells to obtain energy [44]. This switch in immune cells favors anabolic reactions such as the biosynthesis of fatty acids, proteins, and nucleotides in order to provide rapid growth and cell division. Such immune cells include lymphocytes, both Helper (CD4+) and Cytotoxic (CD8+) (CTLs); neutrophil granulocytes; activated macrophages; and NK cells [45]. All of the above cell types play a crucial role in mediating and regulating an efficient host immune response against cancer cells. In order to accomplish their functions, immune cells resident in the tumor micro-environment (TME) compete with cancer cells for the present nutrients. Therefore, the proliferation of cancer cells deprives TME of the nutrients needed by immune cells to carry out their functions.

In addition, immunosuppression is mediated by the induction of the Warburg effect. The prevalence of glycolysis in tumor cells and immune cells causes an increased production of pyruvic acid. The latter is not converted into Acetyl-Coenzyme A, but it is transformed into lactate and is secreted into the extracellular space. This phenomenon significantly increases the acidity of TME, lowering is PH. The latter is extensively reported as an immunosuppressive mechanism [46,47].

Furthermore, most solid tumors, although characterized by increased heterogeneity, exhibits hypoxia conditions compared with normal tissues. Hypoxia influences the immune function through disruption or alteration of the metabolism in TME, infiltrating immune cells. This mechanism is mainly regulated by HIF-1 activity [48].

Furthermore, hypoxia is well known to increase FOXP3 transcription factor levels. FOXP3 is a potent regulator of T-regulatory (T-Reg) lymphocytes. T-Reg lymphocytes display an immunosuppressive activity as they dampen the function of both T-helper and CTLs. Notably, hypoxia favors the production of TGF-b and CCL28, both of which stimulate the T-reg cell activity [49].

In conclusion, acidosis and hypoxia, often associated with tumors, favor the growth of immune suppressive cells more than cancer immune efficient cells. T-Reg lymphocytes and M2 macrophages, in fact, are often up-regulated in tumor infiltrates of hypoxic tumors. Both cell types display immunosuppressive functions through the production of immune suppressive cytokines, such as IL-10 and TGF-Beta. In SCCHN (and especially in tumors associated to smoke and alcohol consumption), a poor immune infiltrate is present, which is most likely represented by a high number of T-reg and low number of CD8+ and CD4+ lymphocytes [10,11,42].

## 5. How Can We Target the Immune Metabolism in SCCHNs

SCCHNs include high heterogeneous subtypes of tumors that differ in both their genetic and biological characteristics. This issue was already extensively highlighted several years ago, when Chung et al. using a microarray technique (upon 582 genes), identified four distinct subtypes of SCCHN: basal-like, classical, mesenchymal, and atypical variants. The basal-like variant displayed a well differentiated histology and showed a high level of expression of EGFR and VEGFR. The mesenchymal subtype was related to the epithelial to mesenchymal transition (EMT) of the cancer cells, showing markers of mesenchymal cells. The atypical variant appeared to differ from the basal-like, displaying poorly differentiated histology and expressing a low level of EGFR. Lastly, the classical variant was related to alcohol and tobacco consumption, showing high levels of antioxidant enzymes, such as Glutathione-S-transferase and thioredoxin reductase [50]. Similar results were reported by Walter et al. In addition, these authors, by including the HPV status in the analysis and assessing the expression of genes such as PIK3CA, showed that HPV-positive tumors were included in the atypical variant, and that this variant often showed the mutation of PI3KCA [51]. At that time, the results could not be translated into the implementation of novel therapeutic strategies, since no drugs targeting these types of alteration were available. Today, novel potential therapeutic opportunities targeting these alterations are available and are currently being investigated in order to interfere with cancer progression, including metabolism reprogramming. In addition, currently, immunotherapy with check-point inhibitors is revolutionizing the clinical approach of SCCHNs. Targeted agents are currently being investigated in combination with immunotherapy. Agents targeting metabolic reprogramming pathway components in combination with immune check-point inhibitors have the potential to counteract cancer progression and immune suppression induced by the metabolic switch in many tumors, including SCCHNs.

### 5.1. HPV Related SCCHN and PI3K Pathway

The results of the studies conducted by Chung and Walter demonstrated the existence of an atypical SCCHN phenotype, often associated with HPV infection, often characterized by a disruption of the PI3K−Akt−mTOR pathway [10,11]. From a careful analysis of the mechanisms underlying metabolic reprogramming, the PI3K−Akt−mTOR pathway is often responsible for a large part of the metabolic modification. As previously discussed, mutations in this pathway accelerate glycolysis; glucose uptake; and anabolism of proteins, fatty acids, and nucleotides, thus favoring rapidly proliferating cells. Several PI3K inhibitors are currently being investigated in solid tumors, including SCCHN. Buparlisib, an oral small molecule and pan-PI3K inhibitor, is currently in an advanced stage of clinical testing and is being investigated in a phase 3 trial in SCCHN patients with recurrent/metastatic squamous tumors following progression to a platinum-containing regimen, as well as to check-point inhibitor-based immunotherapy [52]. Promising results of buparlisib in phase 1 and 2 trials of patients with advanced SCCHN have already been reported [53,54]. In the largest of these two trials, Soulières et al. [54] enrolled 158 patients with recurrent/metastatic platinum-resistant SCCHN. Patients were randomly assigned to receive either buparlisib plus paclitaxel or placebo plus paclitaxel. As a result, the median progression-free survival was 4.6 months in the buparlisib group and 3.5 months in the placebo group (*p* = 0.011). Based on its improved clinical efficacy and manageable safety profile, buparlisib in combination with paclitaxel was demonstrated to be effective in patients with recurrent/metastatic platinum-refractory SCCHN. Rationally, a single drug would be able to act on the right phenotype of SCCHN (HPV-related), and be able to block, at least in part, the metabolic reprogramming mechanisms (Table 1).

### 5.2. Mutagens Related SCCHN (Alcohol and Tobacco)

The results of the studies by Chung and Walter also highlighted the existence of a particular phenotype of SCCHN called “classic”. A crucial feature of this type of SCCHN is the exposure to mutagens and, above all, the high expression of “anti-oxidant” enzymes such as glutathione reductase. As previously described, upregulation of the PPP is one of the hallmarks of metabolic reprogramming. When glycogen is decomposed, a large amount of glucose 6-phosphate flows to the PPP. PPP is of particular importance for tumor cells because it provides the NADPH needed for de novo fatty acid synthesis, increases the ratio of NADPH/NADP+, and promotes the production of Glutathione [9,10,11]. In addition, PPP induces NADPH oxidase (NOX) activation. NOX uses NADPH to reduce oxygen and generate superoxide (O2). SCCHN tumor cells and especially those with the classical phenotype, due to their genetic structure, upregulate the biosynthesis of the fatty acids needed for the biosynthesis of membranes, as well as to detoxify their cytosol from ROS. The latter ability protects cancer cells from ROS-induced apoptosis, increasing their radio and chemoresistance [61]. Drugs selectively targeting this type of SCCHN may be represented, rather than PI3K inhibitors, by some types of non-steroidal anti-inflammatory drugs (NSAIDs). In both clinical and pre-clinical models, NSAIDs have been demonstrated to inhibit cyclo-oxygenase (COX) activity, as well as G6PDH and other NADPH-producing enzymes of PPP, such as 6-phosphogluconate dehydrogenase (6PGD). As a result, NSAIDs are shown to induce apoptosis of cancer cells by oxidative stress, mediated by the depletion of glutathione and NADPH [62,63].

Moreover, the classic-like phenotype, according to the Chung and Walter analysis, often shows CCND1 mutations. CCND1 is an oncogene whose product is Cyclin D1, an enzyme crucial for cell cycle progression. Specifically, Cyclin D1 in association with CDK4/6 acts as a mitogenic sensor of cell cycle progression. CCND1 can be targeted with several drugs including palbociclib, a small molecule CDK4/6 inhibitor currently investigated as a treatment in several solid tumors. In SCCHN, there are phase 2 trials that are currently investigating the use of palbociclib in HPV-negative SCCHN. The latter phenotype strongly resembles the classic subtype SCCHNs from a genetic point of view (Table 1) [56,57,58,59]. A step forward, especially for mutagens related to SCCHN (exhibiting the “classic phenotype”), may be to associate Cyclin D1/ CDK4/6 inhibitors with NSAIDs in order to enhance both metabolism reprogramming and cell cycle progression inhibition.

### 5.3. Drugs with Direct Effect on Metabolism

Metformin, the most widely used drug for type 2 diabetes, might affect cancer aggressiveness. Its mechanism of action consists of inhibiting liver gluconeogenesis and increasing the peripheral uptake of glucose. Lately, newer functions of metformin have been acknowledged. Metformin negatively regulates the PI3K−Akt−mTOR pathway, as a result affecting metabolism reprogramming [64]. Metformin can block mTORC1 and TSC2, thus negatively regulating the small GTP-binding protein RHEB and eventually decreasing mRNA translation and ribosome biogenesis [64]. Moreover, metformin can directly act on glycolysis by blocking the crucial enzyme hexokinases (HKs) that are responsible for the availability of G6P (glucose 6 phosphate). Interestingly, metformin seems to bind the respiratory Complex I and thus inhibit OxoPhos. This latter feature appears to be in contrast with what we have reported. However, some lines of evidence demonstrate that cancer cells not use only glycolysis, but they are able to adapt themselves to the oxygenation status of TME and to also perform OxoPhos when it is necessary [64,65,66]. Therefore, metformin might inhibit both glycolysis and OxoPhos, acting in an anti-proliferative manner, regardless of the oxygenation conditions of TME. Currently, metformin is in a very early phase of experimentation in SCCHN.

## 6. Discussion

The etiology of SCCHNs is very heterogeneous. We can assert that the “primum movens” is represented by crucial DNA mutations, capable of altering the regulatory mechanisms of the cell cycle. As mutations accumulate, the tumor cells become “transformed” and are capable of forming cancer. The process of “neoplastic progression” is, however, crucial, as it can give cancer cells different abilities, which ultimately allow tumor cells to invade organisms. Cell cycle dysregulation is only part of the whole process, which allows cancer cells to acquire additional capabilities such as the evasion of immune surveillance and metabolic reprogramming.

It is well-known that chemotherapy with anti-neoplastic drugs, as well as targeted therapy (which represents a more “targeted” evolution of therapy), interfere with cell cycle progression, counteract the ability of cancer cells to proliferate uncontrollably, and the ability of cancer cells to become “immortal” (as a consequence of the alteration of the mechanisms that lead to apoptosis). These strategies, if used alone, are unlikely to be able to fight a well-developed cancer that has acquired various skills (not only uncontrolled growth). A step forward was made by introducing immunotherapy in the clinical armamentarium. Immunotherapy is a strategy capable of strengthening the immune system, thus making it capable of rejecting cancer cells, taking advantage of the presence of the so-called tumor associated antigens, which are recognized as being foreign by the immune system. As we have discussed, the immune response in the presence of cancer cells is strongly compromised, mainly due to alterations to the HLA class I antigen expression and through the production of immunosuppressive cytokines either directly or indirectly released by the tumor cells.

Desrichard et al. in a pre-clinic/clinic analysis, demonstrated that SCCHNs related to smoking and alcohol have a very low lymphocyte infiltration. This infiltration consists mainly of T-Reg and minimally of CTLs and NK cells. However, in SCCHNs, there is a strike impairment even when the tumor is infiltrated by lymphocytes. Desrichard et al. also demonstrated a strong correlation between high alcohol and tobacco use and a poor response to immunotherapy in SCCHNs [67]. This feature highlights that SCCHNs are often tumors characterized by a strong immune suppression, so therapeutic strategies should be based on drugs that are able to block the immunosuppressive stimuli (produced by tumor cells).

The addition of a “new generation” of check-point inhibitor based-immunotherapy has significantly improved the prognosis of patients with recurrent metastatic SCCHN, both in platinum-sensitive and platinum-refractory patients. However, a fair percentage of patients are unresponsive to immunotherapy, even if associated with chemotherapy, and for those patients, new therapeutic approaches are urgently needed.

It is hypothesized that SCCHNs related to mutagen exposure (alcohol and tobacco) characterized by a low immune infiltrate often create an immune suppressive microenvironment that can be further damaged by the metabolic shift. SCCHNs showing the “classic phenotype” are characterized by a high expression of detoxifying enzymes, suggesting a hyper-activation of the PPP and of the Warburg effect. Similarly, HPV-related SCCHNs often show PI3KCA activating mutations, which are related to the over activation of glycolysis (Warburg effect), PPP, biosynthesis of fatty acids, nucleotides, and proteins. We could thus speculate that SCCHNs, independently from their genetics and biology (both mutagens and virus related), are always characterized by different grades of immune suppression and mainly by metabolic reprogramming. So, strategies able to combine immunotherapy, targeted therapy, and metabolic targeting therapies should be considered.

As already mentioned, for some SCCHNs, there is already the possibility to act on cell cycle, immune evasion, and metabolism. Approaches that interfere with the metabolic reprogramming of cancer cells can easily “coexist” with targeted therapy, chemotherapy, and immunotherapy. Sometimes, targeted therapy “coincides” with the blocking of metabolic reprogramming, as demonstrated by the PI3K inhibitor application. In addition, metabolic reprogramming can dampen the immune response, modifying the TME. As a result, the implementation of strategies combining check-point inhibitors and drugs targeting metabolic reprogramming might restore the immune response.

## 7. Conclusions

The process of neoplastic progression involves the acquisition of different abilities by tumor cells, some of which concern dysregulation of the cell cycle, while others concern the ability to evade immune response, and others concern the ability to modify the metabolism in a way that allows for obtaining an advantage in survival.

An accurate approach aimed at fighting SCCHNs should consider perfect knowledge of their genetics, which in turn could help to employ the correct drugs that are able to interfere with the disrupted pathways. An ideal systemic therapy should aim to “block” all of the skills acquired by tumor cells during neoplastic progression, thus acting on the cell cycle, immune evasion, and metabolic reprogramming (Figure 3).

## Figures and Tables

**Figure 1 cancers-14-03560-f001:**
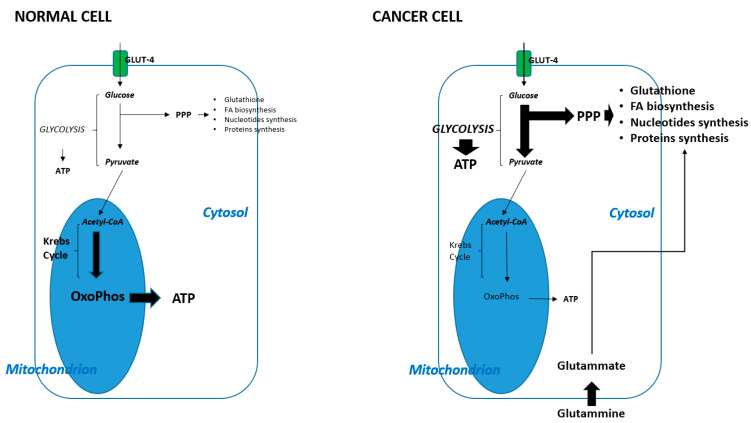
Distinctive features between metabolism in normal and cancer cells. The greater thickness of the arrows indicates the greater frequency of metabolic pathway occurring. PPP: pentose phosphate pathway; OxoPhos: oxidative phosphorylation; FA: fatty acids.

**Figure 2 cancers-14-03560-f002:**
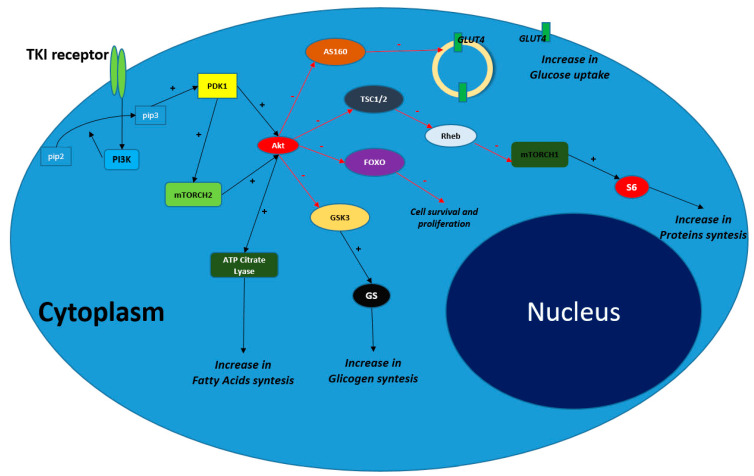
Intracellular down-stream pathways stimulated by PI3K-Akt-mTOR activation. The direct consequences are an (1) increase in glucose uptake through GLUT4 upregulation; (2) increase in protein synthesis through S6 (ribosomal subunits) up-regulation; (3) increase in glycogen synthesis (which is available thus for the glycolysis); (4) increase in fatty acids synthesis through ATP Citrate lyase up-regulation and ultimately (5) block of FOXO which acts as tumor suppressor genes, increasing cell proliferation. PI3K: Phosphatidyl Inositol 3 Kinase; pip2: Phosphatidyl Inositol 2-phosphate; pip3: Phosphatidyl Inositole3 phosphate; PDK1: phosphoinositide dependent kinase-1, mTORCH: mammalian target of rapamycin complex; AS160: Akt substrate of 160 kDa; TSC 1/2; hamartin–tuberin complex; foxo: Fork head box O3; GSK3: Glycogen synthase kinase 3; GS: Glycogen synthase; Rheb: Ras homologue enriched in brain.

**Figure 3 cancers-14-03560-f003:**
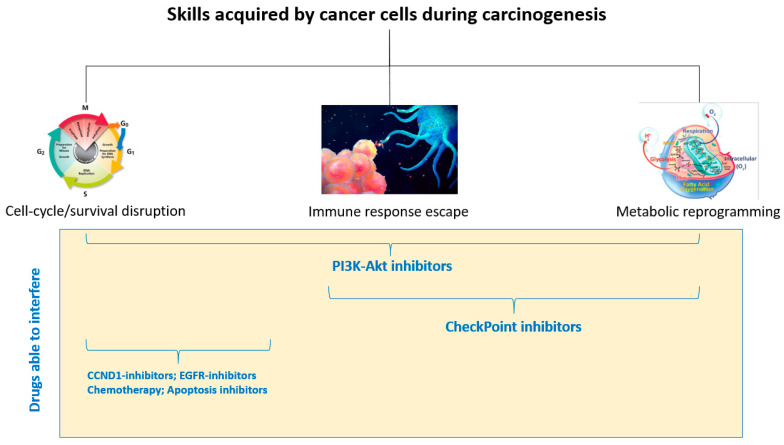
During cancer progression, cancer cells acquire various skills, in particular the ability to alter the cell cycle/survival, the ability to evade the immune response, and the ability to reprogram their own metabolism. An ideal clinical approach could use drugs capable of acting on one or more of the aforementioned skills.

**Table 1 cancers-14-03560-t001:** Drugs targeting SCCHN based on their driver mutation, which are also able to interfere with tumor metabolism (ended clinical trials).

STUDY	Phase	Design (Number of p.ts)	Results	Type of Drug
Head Neck. 2019 Nov.;41(11):3842–3849 [53].	Phase I	Buparlisib + Cetuximab (12)	Safe at a dose of 100 mg/daily	Anti-PI3K
Lancet Oncol. 2017 Mar.;18(3):323–335 [54].	Phase II	Buparlisib + Paclitaxel (158)	Median progression-free survival was 4.6 months (endpoint met)	Anti PI3K
Invest New Drugs. 2021 Dec.;39(6):1641–1648 [55].	Phase I	Copanlisib + Cetuximab (11)	Safe at a dose of 30 mg/daily	Anti PI3K
Oral Oncol. 2021 Apr.;115:105192 [56].	Phase II	Palbociclib +/− Cetuximab (125)	No differences in OS	Anti CDK 4/6
Oral Oncol. 2021 Mar.;114:105164 [57].	Phase II	Palbociclib + Cetuximab (24)	ORR: 4% (endpoint not met)	Anti CDK 4/6
Invest New Drugs. 2020 Oct.;38(5):1550–1558 [58].	Phase II	CBDCA + Palbociclib (21)	DCR: 23% (endpoint not met)	Anti CDK 4/6
Lancet Oncol. 2019 Sep.;20(9):1295–1305 [59].	Phase II	Palbociclib + Cetuximab (62)	ORR 39% (endpoint met)	Anti CDK 4/6
Oral Oncol. 2016 Jul.;58:41–48 [60].	Phase I	Palbociclib + Cetuximab (9)	Safe at a dose of 125 mg/daily	Anti CDK 4/6

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
