# Peer review of "Cancer Cell Metabolism Reprogramming and Its Potential Implications on Therapy in Squamous Cell Carcinoma of the Head and Neck: A Review"

_cancers, 2022, doi:10.3390/cancers14153560_

Round 1

Reviewer 1 Report

  The manuscript entitled “Cancer cells metabolism reprogramming and its possible implication on therapy in Squamous cell carcinoma of the head and neck: a review” by Caponigro F et al., has presented an overview of the metabolism reprogramming of cancer cells, the immune system, and the interplay between metabolic shift and immune response. Overall, the manuscript is well organized with comprehensive elaborations. However, some issues should be corrected before the consideration of publishment in Cancers.  

1.     Despite the drugs that have been used in preclinical/clinical studies, the newest therapeutic regimen/synthesized drugs that are reported to alter the cancer cell metabolism should be introduced or outlined in the manuscript. In this case, more figures/schemes from the cited papers should be involved here to make the manuscript more interesting and enjoyable to read.

2.     The scheme in this manuscript is poorly drawn, what does the dark blue circle mean?

3.     The language of the manuscript should be formally used. The authors should avoid using spoken languages, such as “very”, and “This latter appears to be in contrast to what said before”.

4.     Typos such as “above all energy gain. and macromilecules” in the Abstract should be avoided.

5.     The vagueness of the sentence throughout the manuscript should be corrected, such as “In conclusion, the acidosis and hypoxia that is often associated with it, favors the growth of some cells of the immune system over others. 

Based on the comments above, this manuscript is suggested to have a major revision before the consideration of acceptance.

Author Response

  • Despite the drugs that have been used in preclinical/clinical studies, the newest therapeutic regimen/synthesized drugs that are reported to alter the cancer cell metabolism should be introduced or outlined in the manuscript. In this case, more figures/schemes from the cited papers should be involved here to make the manuscript more interesting and enjoyable to read.

Response: in accordance with this suggestion, we have added another figure in the text and we have better described one of the most important clinical studies concerning the issue (the one that describes the activity and efficacy of PI3K inhibitors

  • The scheme in this manuscript is poorly drawn, what does the dark blue circle mean?

Response: we have corrected the mistakes

  • The language of the manuscript should be formally used. The authors should avoid using spoken languages, such as “very”, and “This latter appears to be in contrast to what said before”.

Response: we have deleted these statements

  • Typos such as “above all energy gain. and macromilecules” in the Abstract should be avoided.

Response: we have corrected the errors

  • The vagueness of the sentence throughout the manuscript should be corrected, such as “In conclusion, the acidosis and hypoxia that is often associated with it, favors the growth of some cells of the immune system over others

Response: we have corrected these sentences, better specifying their significance in the text

Reviewer 2 Report

In this review the authors describe the metabolic reprogramming during the acquisition of the cancerous phenotype. As reported, the transition of normal cell to cancer cell is characterized by several biological events, which often leads to the invasion of the whole organism by cancer cells and metastases. Although the theme of this review is very interesting, there are several major concerns that impede a complete understanding of the text. Apart from the English, which must necessarily be improved, all the paragraphs are described as a "list of events" that occur during the carcinogenic transition. In my opinion, it would be more interesting if the authors focused more on describing the scientific result that involve the mutated pathways they have listed. 

Moreover, it would be advisable to add other figures that highlight the importance of some results (e.g. figures of some cited papers, ask permission).

Finally, the authors should focus more on the main message of this review and emphasize more what is its novelty.

Author Response

  • In this review the authors describe the metabolic reprogramming during the acquisition of the cancerous phenotype. As reported, the transition of normal cell to cancer cell is characterized by several biological events, which often leads to the invasion of the whole organism by cancer cells and metastases. Although the theme of this review is very interesting, there are several major concerns that impede a complete understanding of the text. Apart from the English, which must necessarily be improved, all the paragraphs are described as a "list of events" that occur during the carcinogenic transition. In my opinion, it would be more interesting if the authors focused more on describing the scientific result that involve the mutated pathways they have listed

Response: First of all, we have given the manuscript to an English speaker for the appropriate corrections, as recommended by you.

Moreover, in accordance with your suggestion, we have further specified in the text (adding in some case an appropriate bibliography) the mechanisms underlying the metabolic reprogramming that mostly involve TP53 and PI3KCA. However, we preferred not to go too far (adding further pre-clinical studies) so as not to affect the reader's understanding and so as not to further lengthen the text.

  • Moreover, it would be advisable to add other figures that highlight the importance of some results (e.g. figures of some cited papers, ask permission).

Response: thanks a lot for the suggestion. We have added another figures for better illustrating the metabolic reprogramming and we absolutely confirm that none of the figures / tables included in the text have been taken elsewhere without permission, but they are of our production.

  • Finally, the authors should focus more on the main message of this review and emphasize more what is its novelty

Response: Thanks for the statement. in accordance with your suggestion we have further specified in several points in the text, and in particular in the concluding lines, that the "take home message" is the future integration of chemotherapy, targeted therapy, immunotherapy and strategies acting on the metabolic reprogramming in SCCHNs

Reviewer 3 Report

Perri et al., aimed to write a review about the metabolic reprogramming in SCCHN. Unfortunally, the paper lacks a couple components regarding this subject. Throughout the paper either the citations hardly refer to SCCHN even when they to the implications except of paragraph 5.1 are minimum.
Next to this important citations are missing and parts of the text are not supported by references.

Also the authors missed the opportunity to explain what makes SCCHN tumors different than other solid tumors in regard to metabolic changes. What are the main implications on treatment etc, although sometimes it is briefly mentioned the insights are missing. The contribution to the field is therefore lacking.
Additionally there are a lot of spelling/language/text mistakes. see e.g. the following line on page 4: On the other hand, the majority of TP53 mutations are "missense", namely capable of giving a functioning protein, which nevertheless ac-quires a function (different from that of the wild type protein).  It should state that is incapable of giving a functional protein/... which is exactly the opposite. 

Author Response

  • Perri et al., aimed to write a review about the metabolic reprogramming in SCCHN. Unfortunally, the paper lacks a couple components regarding this subject. Throughout the paper either the citations hardly refer to SCCHN even when they to the implications except of paragraph 5.1 are minimum.

Response: Thanks a lot for your suggestion. In accordance to your recommendations, we have added specific statements on SCCHNs throughout the text, further underlining that the pathways involved in metabolic reprogramming are often dysregulated particularly in SCCHNs (e.g. TP53 and PI3KCA)

  • Next to this important citations are missing and parts of the text are not supported by references. Also the authors missed the opportunity to explain what makes SCCHN tumors different than other solid tumors in regard to metabolic changes. What are the main implications on treatment etc, although sometimes it is briefly mentioned the insights are missing. The contribution to the field is therefore lacking.

Responses: we agree with these suggestions. Thus, we have further specified in the text (supporting it with bibliographic notes in the right places) the mechanisms underlying metabolic reprogramming in particular in SCCHNs (mutations of TP53, PI3KCA, HIF, PTEN). In other words, we explained that some mutations responsible for the dysregulation of these pathways are very frequent in SCCHNs. We finally clarified and underlined (see final paragraph) that the "take home message" is the future integration of chemotherapy, targeted therapy, immunotherapy and strategies acting on metabolic reprogramming in SCCHNs. We believe that the future contribution in the field may be significant, as some therapeutic strategies (see block of the PI3K-Akt pathway) can act both on the cell cycle and on metabolic reprogramming and that the integration of these strategies with immunotherapy can be taken into consideration. In general, translational research can be the starting point for designing strategies based on the genetics of SCCHNs.

  • Additionally there are a lot of spelling/language/text mistakes. see e.g. the following line on page 4: On the other hand, the majority of TP53 mutations are "missense", namely capable of giving a functioning protein, which nevertheless ac-quires a function (different from that of the wild type protein). It should state that is incapable of giving a functional protein/... which is exactly the opposite.

Response: We agree with the review. We have corrected the inaccuracies and we specified better that the mutant protein (P53), deriving from mutations that do not "truncate" the product, is however capable of performing functions which in this case are the opposite ones with respect to the wild type protein. In the text there is the sentence: “In the case of TP53 missense mutations, the result is a protein that performs opposite functions to those performed by the wild type protein”. Moreover and mainly, we have given the manuscript to an English speaker for the appropriate corrections, as recommended by you.

Round 2

Reviewer 1 Report

I suggest that this manuscript can be accepted after the correction of typos in Figure 2.

Author Response

Response rev 1

S: I suggest that this manuscript can be accepted after the correction of typos in Figure 2.

Res: Done

Reviewer 2 Report

After major revisions the manuscript has improved considerably. However, there are still some concerns in the text that need to be revised, mainly with regard of the English. Below is a list of some sentences to edit as suggested: 

Add a ref to ". Specific types of cells of the immune system - cytotoxic T lymphocytes and Natural Killer lymphocytes- accomplish this task”

Paragraph 2. Edit the period “Rapidly proliferating cells, including cancer cells, thus need, in addition to the immediate availability of energy (in the form of ATP), also, and above all, to regenerate the plasma membranes that must be duplicated during mitosis and to increase the synthesis of proteins and nucleotides (necessary for DNA duplication).” As follow: “Thus, in addition to the immediate availability of energy, the rapidly proliferating cells need to regenerate the plasma membranes that must be duplicated during mitosis and to increase the synthesis of proteins and nucleotides (necessary for DNA duplication).”

Edit this sentence “In addition, in order to maintain energy homeostasis, cancer cells can consume not only large amounts of glucose, but also another key metabolite, namely glutamine, which can be considered as the second principal growth supporting substrate.” Remove the word “namely” and replace with “the" glutamine.

Edit this sentence “In conclusion, cancer cells often, but not always, show a metabolic shift that favors some reactions rather than others and all this is strictly functional to their need to grow and divide rapidly. The above mentioned features are described in figure 1.” Please remove “but not always”.

 Edit this sentence “M2 macrophages , thus suppress the host anti-cancer immune response, leading to tumor progression” as follow: “Thus, M2 macrophages suppress the host anti-cancer immune response, leading to tumor progression”

paragraph 3:

Please edit this sentence “Well understood major mechanisms,….” As follow : The well-established major mechanisms,….”

Paragraph 3.3

Edit “Hypoxia inducible factor-1alpha (HIF-1alpha) is a transcription factor which, under normal oxygenation conditions….” Replace “which” with “that”

Edit this sentence “In hypoxic conditions it is instead stabilized and then upregulated and, by binding to promoter sequences on the DNA, it stimulates the transcription of genes involved in angiogenesis and anaerobic glycolysis.” As follow: Instead, In hypoxic conditions it is stabilized and then upregulated. Moreover, HIF-1alpha stimulates the transcription of genes involved in angiogenesis and anaerobic glycolysis by binding the promoter sequences on the DNA”

Edit this sentence “Only in some cases the hyper expression of HIF-1alpha is due to gene mutations with "gain of function”………” replace the word “hyper expression” with “upregulation”

Edit this sentence “The result in both cases is an up regulation of glycolysis and therefore of the Warburg effect.” Replace the word upregulation with “hyper activation”

Edit this sentence “PTEN is a natural inhibitor of PI3K, so loss-of-function mutations induce the hyper-activation of the Akt pathway and all the downstream effectors of the pathway.” As follow: PTEN is a natural inhibitor of PI3K and the loss-of-function mutations induce the hyper-activation of the Akt and its the downstream effectors”

Paragraph 4.2

Edit this sentence: “…..as often reported in scientific literature.” Please remove the word “scientific”

Please rephrase this sentence “Most solid tumors, although heterogeneous, exhibit hypoxia conditions if compared to normal tissues, and hypoxia influences the immune function through disrupting or altering metabolism in immune cells infiltrated in TME”. Please improve the English.

Paragraph 5

Please rephrase this sentence “Now we are able to integrate the results obtained at that time with the therapeutic opportunities now available able to interfere with important and frequently disrupted genetic pathways” It is not clear. 

In this sentences “s. In addition, we can consider the so-called "new generation" immunotherapy, which can be applied in clinical practice, especially in association with targeted therapy. Finally we could integrate targeted therapies with immunotherapy and mainly with therapies able to disrupt the metabolic shift in SCCHNs” why do authors use “we”? please rephrase.

Paragraph 6

Please rephrase this sentence “These strategies, therefore, are unlikely to be able, if used alone, to fight a well-developed cancer, which has therefore acquired various skills (not only uncontrolled growth).” 

Please rephrase “In addition, metabolic reprogramming does dampen the immune response, modifying the TME, so using check-point inhibitors and drugs acting on metabolic reprogramming may also restore the immune-response.” What does the authors mean with “does dampen”?

Please rephrase “In the near future, the ideal systemic therapy should aim to "block" all the skills acquired by tumor cells during the phenomenon of neoplastic progression, thus acting on the cell cycle, on the immune-surveillance and on the metabolic reprogramming” please edit as follow : In the near future, the ideal systemic therapy should aim to "block" all the skills acquired by tumor cells during the phenomenon of neoplastic progression, thus acting on the cell cycle, the immune-surveillance and the metabolic reprogramming

Figure 21: Improve the english

In general, a further improvement of the English in the entire manuscript is needed. 

Author Response

Response rev 2

S: Add a ref to ". Specific types of cells of the immune system - cytotoxic T lymphocytes and Natural Killer lymphocytes- accomplish this task”

Res: Done

S: Paragraph 2. Edit the period “Rapidly proliferating cells, including cancer cells, thus need, in addition to the immediate availability of energy (in the form of ATP), also, and above all, to regenerate the plasma membranes that must be duplicated during mitosis and to increase the synthesis of proteins and nucleotides (necessary for DNA duplication).” As follow: “Thus, in addition to the immediate availability of energy, the rapidly proliferating cells need to regenerate the plasma membranes that must be duplicated during mitosis and to increase the synthesis of proteins and nucleotides (necessary for DNA duplication).”

Res: We asked for the help of an English speaker who changed the sentence

S:Edit this sentence “In addition, in order to maintain energy homeostasis, cancer cells can consume not only large amounts of glucose, but also another key metabolite, namely glutamine, which can be considered as the second principal growth supporting substrate.” Remove the word “namely” and replace with “the" glutamine.

Res: We asked for the help of an English speaker who changed the sentence

S: Edit this sentence “In conclusion, cancer cells often, but not always, show a metabolic shift that favors some reactions rather than others and all this is strictly functional to their need to grow and divide rapidly. The above mentioned features are described in figure 1.” Please remove “but not always”.

Res: We asked for the help of an English speaker who changed the sentence

S:  Edit this sentence “M2 macrophages , thus suppress the host anti-cancer immune response, leading to tumor progression” as follow: “Thus, M2 macrophages suppress the host anti-cancer immune response, leading to tumor progression”

Res: We asked for the help of an English speaker who changed the sentence as you suggested

S: Please edit this sentence “Well understood major mechanisms,….” As follow : The well-established major mechanisms,….”

Res: Done

S: Edit “Hypoxia inducible factor-1alpha (HIF-1alpha) is a transcription factor which, under normal oxygenation conditions….” Replace “which” with “that”

Res: We asked for the help of an English speaker who changed the sentence

S: Edit this sentence “In hypoxic conditions it is instead stabilized and then upregulated and, by binding to promoter sequences on the DNA, it stimulates the transcription of genes involved in angiogenesis and anaerobic glycolysis.” As follow: Instead, In hypoxic conditions it is stabilized and then upregulated. Moreover, HIF-1alpha stimulates the transcription of genes involved in angiogenesis and anaerobic glycolysis by binding the promoter sequences on the DNA”

Res: We asked for the help of an English speaker who changed the sentence as you suggested

S: Edit this sentence “Only in some cases the hyper expression of HIF-1alpha is due to gene mutations with "gain of function”………” replace the word “hyper expression” with “upregulation”

Res: Done

S: Edit this sentence “The result in both cases is an up regulation of glycolysis and therefore of the Warburg effect.” Replace the word upregulation with “hyper activation”

Res: Done

S: Edit this sentence “PTEN is a natural inhibitor of PI3K, so loss-of-function mutations induce the hyper-activation of the Akt pathway and all the downstream effectors of the pathway.” As follow: PTEN is a natural inhibitor of PI3K and the loss-of-function mutations induce the hyper-activation of the Akt and its the downstream effectors”

Res: We asked for the help of an English speaker who changed the sentence in a similar way to what you suggested

S: Edit this sentence: “…..as often reported in scientific literature.” Please remove the word “scientific”

Res: Done

S: Please rephrase this sentence “Most solid tumors, although heterogeneous, exhibit hypoxia conditions if compared to normal tissues, and hypoxia influences the immune function through disrupting or altering metabolism in immune cells infiltrated in TME”. Please improve the English.

Res:

S: Please rephrase this sentence “Now we are able to integrate the results obtained at that time with the therapeutic opportunities now available able to interfere with important and frequently disrupted genetic pathways” It is not clear. 

Res: We asked for the help of an English speaker who changed the sentence in a similar way to what you suggested

S: In this sentences “s. In addition, we can consider the so-called "new generation" immunotherapy, which can be applied in clinical practice, especially in association with targeted therapy. Finally we could integrate targeted therapies with immunotherapy and mainly with therapies able to disrupt the metabolic shift in SCCHNs” why do authors use “we”? please rephrase.

Res: We asked for the help of an English speaker who changed the sentence in a similar way to what you suggested

S: Please rephrase this sentence “These strategies, therefore, are unlikely to be able, if used alone, to fight a well-developed cancer, which has therefore acquired various skills (not only uncontrolled growth).” 

Res: We asked for the help of an English speaker who changed the sentence in a similar way to what you suggested

S: Please rephrase “In addition, metabolic reprogramming does dampen the immune response, modifying the TME, so using check-point inhibitors and drugs acting on metabolic reprogramming may also restore the immune-response.” What does the authors mean with “does dampen”?

Res: We asked for the help of an English speaker who changed the sentence in a similar way to what you suggested

S: Please rephrase “In the near future, the ideal systemic therapy should aim to "block" all the skills acquired by tumor cells during the phenomenon of neoplastic progression, thus acting on the cell cycle, on the immune-surveillance and on the metabolic reprogramming” please edit as follow : In the near future, the ideal systemic therapy should aim to "block" all the skills acquired by tumor cells during the phenomenon of neoplastic progression, thus acting on the cell cycle, the immune-surveillance and the metabolic reprogramming

Res: We asked for the help of an English speaker who changed the sentence in a similar way to what you suggested

S: Figure 21: Improve the english

Res: Done
